# Standing Meetings Are Feasible and Effective in Reducing Sitting Time among Office Workers—Walking Meetings Are Not: Mixed-Methods Results on the Feasibility and Effectiveness of Active Meetings Based on Data from the “Take a Stand!” Study

**DOI:** 10.3390/ijerph17051713

**Published:** 2020-03-05

**Authors:** Ida H Danquah, Janne S Tolstrup

**Affiliations:** National Institute of Public Health, University of Southern Denmark, 1455 Copenhagen, Denmark; idah@niph.dk

**Keywords:** sitting time, workplace, occupational sitting, intervention, sedentary behavior, meetings, active meetings, standing meetings, walking meetings, mixed methods

## Abstract

Active meetings (standing or walking) have the potential to reduce sitting time among office workers. The aim of the present study was to explore the feasibility and effectiveness of standing and walking meetings. The “Take a Stand!” study was a cluster-randomized trial, consisting of multiple components including the possibility of active meetings. Analyses were based on the 173 participants in the intervention group. Feasibility was evaluated by questionnaire and interview data from participants, ambassadors and leaders. Effectiveness was assessed as the change in objectively measured sitting time from baseline to 3 months follow-up. Regular standing meetings were implemented at all offices and were generally popular, as they were perceived as more effective and focused. In contrast, only a few walking meetings were completed, and these were generally associated with several barriers and perceived as ineffective. Participants who participated in standing meetings on a regular basis had 59 min less sitting per 8 h workday (95%CI −101;−17) compared to participants who did not participate in standing meetings at all. Walking meeting participation was not significantly associated with changes in sitting time, likely due to the low number of employees who used this option. This explorative study concludes that standing meetings in office workplaces were feasible and well-liked by the employees, and having frequent standing meetings was associated with reduced sitting time. In contrast, walking meetings were unfeasible and less liked, and thus had no effect on sitting time.

## 1. Introduction

A high level of sitting time is detrimental to health [1,2]. Among office workers, multicomponent interventions, including changes to the office environment, changes in the organization of the work and provision of information, have proved effective to reduce sitting time [3]. One element often included in such interventions is standing and walking meetings. However, only very little research has been done into this area, with studies being pilot studies or focusing on the perceptions of standing and walking meetings [4,5,6,7]. Additionally, a review of sitting time interventions at the workplace by Shrestha et al. [3] concluded that there were no studies specifically investigating the effects of standing or walking meetings on sitting time.

Thus, the purpose of this study was to explore the feasibility of standing and walking meetings at the office workplace and to assess the effectiveness of these active meeting forms on reducing sitting time, using data from the “Take a Stand!” study.

Feasibility included the implementation of the active-meeting component of the “Take a Stand!” intervention, and the views of the participants on enablers and barriers regarding standing and walking meetings. Effectiveness was assessed as the reduction in sitting time during work hours after 3 months, depending on the frequency of standing and walking meetings.

## 2. Materials and Methods

The “Take a Stand!” study was a cluster-randomized trial resulting in effective reductions in sitting time [8,9]. The intervention is described in detail elsewhere [8], but in short, it consisted of five components: (i) Appointment of ambassadors and management support; (ii) Environmental changes, e.g., installation of high meeting tables and definition of routes for walking meetings; (iii) Lecture on sedentary behavior and health; (iv) Workshop to ensure local adaptation at the individual, office and workplace levels through four themes: using a sit-stand desk, breaking up sitting, standing and walking meetings and setting common goals at office level; and (v) Optional weekly e-mails and biweekly text messages. The intervention focused on four strategies to reduce sitting: using a sit-stand desk, breaking up prolonged periods of sitting, having standing and walking meetings, and setting common goals at the office level. Control participants were instructed to behave as usual. Regarding the element on standing and walking meetings, all offices were obliged to provide facilities for standing meetings (both formal in meeting rooms and informal, e.g., on corridors) and walking meetings (suggested routes). During the workshop, a short introduction was given to the different meeting types with tips and information on facilities, before participants discussed which of their meetings were suitable for standing or walking. During the common goal setting at the workshop, most offices decided on a regular meeting to be held standing.

Four workplaces (three public and one private) with four to six offices (clusters) participated in the “Take a Stand!” study, reaching a total of 317 participants from 19 clusters. We included intervention offices only (10 clusters with a total of 173 participants), as the necessary information for the present study was obtained only in this group. Details on inclusion and exclusion criteria, recruitment processes, randomization, and data on the included offices are described elsewhere [8]. In short, eligible individuals had to be ≥18 years old, work >4 days/week, and not have a pregnancy, sickness or disability affecting their ability to stand or walk. All participants had sit-stand desks prior to inclusion. The study was approved by the local Ethics Committees in Denmark (H-6-2013-005) and in Greenland (project 20914-3, id: 2014-095402). Procedures were designed in accordance with the Declaration of Helsinki. The study was prospectively registered at www.clinicaltrials.gov (NCT01996176). Data were collected in 2013–2014 and analyzed in 2019.

### 2.1. Data Collection

Information on the frequency of standing and walking meetings, facilities and background variables was collected in the intervention group by web-based questionnaires at baseline and at 3 months follow-up. Participation was measured as no, occasional, or regular participation in standing and walking meetings, including both scheduled (formal) and unscheduled (informal) meetings. Further details on questionnaire data collection can be found elsewhere [8].

Sitting time was measured by an ActiGraph GT3X+ accelerometer worn on the front of the thigh for 5 days (Monday–Friday) [8,10,11,12]. Interview data were obtained from 11 focus groups with participants and 20 semi-structured interviews with ambassadors and leaders conducted at the 10 intervention offices after the final follow-up at 3 months. One theme during these interviews was the evaluation of the active-meeting component of the intervention.

### 2.2. Data Analysis

Statistical analysis was conducted using STATA/IC-14.0. Multilevel mixed-effects linear regression was used with 3-month sitting time as the outcome, taking baseline sitting, age and sex into account. Variables on participation in standing and walking meetings were tested consecutively. Sensitivity analysis included baseline values of leisure time, physical activity and steps, one by one. Interviews were transcribed verbatim and imported into Nvivo. Directed content analysis was used to group findings into nodes including notes on each intervention component [13]. Coding was done by the first author (I.H.D.).

Integration of methods: Different methods were used to answer different parts of the research question, resulting in integration at the design and reporting level as defined by Fetters et al. [14]. At the study design level, data were collected and analyzed separately in order to answer the different parts of the research question as described below; at the reporting level, a contiguous approach was used, presenting results in different sections.

Feasibility was described using both questionnaire data (knowledge of facilities and frequency of the different meeting types) and information from interviews on, e.g., enablers and barriers towards these meetings. Efficacy was analyzed as the difference in objectively measured sitting time at three months, depending on the frequency of participation in standing and walking meetings as measured by the questionnaire.

## 3. Results

Baseline characteristics of participants are shown in Table 1. The baseline questionnaire was filled in for all 173 participants in the intervention group, while the valid activity measure was available for 161 participants at baseline and 141 at 3 months follow-up. Reasons for loss to follow-up were participants not being present when measurements took place (e.g., due to holiday or sickness), a declining Actigraph, and skin irritation. In addition, seven workers discontinued the intervention after 3 months, because they withdrew from the project (2), left the workplace (4) or were on prolonged leave (1). A full flowchart of the trial is published elsewhere [8].

### 3.1. Feasibility

At 3 months follow-up, 94% (144) and 89% (135) of participants stated knowledge of where to have standing and walking meetings. Furthermore, 92% (132) and 93% (125) found facilities for standing and walking meetings satisfactory (Table 1).

#### 3.1.1. Standing Meetings

Almost all participants (93%) participated in standing meetings regularly or occasionally (Figure 1). At most offices, standing meetings were implemented as a part of the general meeting routine, and facilities (high meeting tables) were essential. If the right facilities were in place, meetings were held standing by default, as one participant said: “As long as the desks are in the up position, so are we.” (Ambassador, C5). However, if high meeting tables were lacking or difficult to adjust to a high position, this was a barrier. In some offices, standing meetings were implemented at low tables; however, this was found to be difficult due to problems with light and writing notes. As one participant explained: “It shouldn’t take all of the blame, but the physical surroundings make it a bit of a hassle to have to stand up and write notes on a table that is down at standard height.” (Focus Group, B14).

Other barriers included the different heights of participants making it difficult to agree on a table position; that it could be harder for the chairperson to oversee standing rather than sitting participants; and difficulties introducing standing meetings to people from the outside, who were unfamiliar with the concept.

On the other hand, several participants described how they felt standing meetings were more effective, and that participants were more focused and attentive and participated more actively. As one participant explained: “In addition, you avoid one or two of your colleagues leaning back and zoning out, like they do when everybody’s seated. In one way or another, you’re more active and engaged, or at least present, when standing up. [...] It changes the attitude at meetings, it makes them different, quite simply.” (Focus Group, C9).

#### 3.1.2. Walking Meetings

Walking meetings were generally not implemented (Figure 1). Several ambassadors described how they, in vain, tried to implement the concept. As one ambassador described: “The idea of walking meetings—I was a real advocate for that and really wanted to do it. I’ve been thinking about an appropriate way to do it, but I’ve found it really hard.” (Ambassador, A11).

Several barriers were mentioned, including the following practical issues: weather; the need to write or look at a computer; difficulty walking and talking if there were more than two participants; and the lack of discretion, as walking meetings take place in public spaces. In addition, participants mentioned that it was generally hard to keep focus while walking. Finally, walking meetings often needed more time for preparation, and in some offices, participants felt that leaving the workplace could be perceived as “skipping” work. However, at two workplaces, part of the annual staff development interviews was successfully done walking.

### 3.2. Effectiveness

Office workers who participated in standing meetings on a regular basis had 59 min less sitting time per 8 h workday after 3 months (Confidence Interval, CI 95% −101; −17) as compared to office workers in the intervention group who did not participate in standing meetings. Participation in walking meetings was not associated with reduced sitting time (Figure 1). Sensitivity analysis, taking baseline physical activity and steps into account, did not change these results (Appendix A).

## 4. Discussion

In office workplaces, standing meetings were feasible and well-liked by the employees, and having frequent standing meetings was associated with reduced sitting time. In contrast, walking meetings were unfeasible and less liked and had no significant effect on sitting time.

Hadgraft et al. [6] interviewed 20 office workers about their thoughts on standing and walking meetings. Several of their results are supported by findings in our study. For example, participants found that standing meetings were generally feasible and led to more effective meetings while highlighting the need for the correct facilities. Furthermore, like we confirmed, they did not find walking meetings feasible.

Studies have tested the implementation of standing meetings with positive evaluations; however, the need for the right facilities was emphasized and low tables were mentioned as a barrier [4,7]. Finally, they found that standing up during meetings might violate some unwritten rules because standing is a power signal. This was also confirmed in our study, where chairpersons found it harder to oversee standing meetings. Thus, in order for standing meetings to be implemented successfully, there is a need for change in the organizational culture around meetings and for organizational support towards a new meeting structure.

The finding that standing meetings are perceived to be more productive than sitting meetings [4,7] is supported by an experimental setting finding that standing meetings were on average 34% shorter, but had the same decision quality [15].

Regarding walking meetings, a 3-week pilot-study with 17 participants yielded positive evaluations of a structured scheme with a single 30-min walking meeting per week [5]. However, a qualitative evaluation of a multicomponent intervention among 129 office workers found walk-talk meetings and lunchtime walking groups to be the least used strategies [16]. Reasons for this included time pressure and cultural norms, which is supported by the present study. Practical barriers such as the weather, need for a computer and lack of discretion in walking meetings were found both in our study and in the study by Wahlström et al. [17], who included walking meetings as one component among several in their physical activity promotion program.

### Strengths and Limitations

Strengths of the present study include the mixed-methods design that combines questionnaire data on the use of the different meeting types with interview data on the feasibility and activity data on the effects on sitting time, giving insights into both sitting time effects and participants’ perceptions. Other strengths include the fact that we have interviews with both employees and ambassadors and leaders, the number of participants with accelerometer and questionnaire data, and the objectively measured sitting time with the accelerometer on the thigh.

Limitations include the lack of a control group and the fact that participation in standing and walking meetings was measured by a simple self-report question discriminating between no, occasional and regular participation; however, it is unclear how participants understood the terms ‘occasional’ and ‘regular’, and thus we do not know the exact frequency. In addition, it is unknown exactly how long the meetings lasted; however, the included standing meetings were standing-only meetings, as another question considered meetings with standing or walking elements.

Furthermore, we cannot tell whether the lower sitting time among standing meeting participants was due to less time spent sitting in meetings, or combined with increased use of their sit-stand desk. As the reduction was 59 min per day, it is reasonable to assume a combination of reduced sitting at meetings and during the rest of the day. It might be that standing meetings contribute to the general awareness of the project and the social support between participants, both of which contribute to reduced sitting during the rest of the day. Thus, more research is needed on the feasibility and effects of different types of standing meetings and how standing meetings affect sitting while performing other tasks during the workday, by including more detailed questions on meetings, participant observations or interviews focusing on meetings.

Finally, participation in walking meetings was very limited. However, during other circumstances, walking meetings might be more feasible and thus contribute to a reduction in sitting time.

Based on our findings, we suggest standing meetings to be a feasible and effective strategy to reduce sitting time among office workers. We suggest future studies to consider how the organizational culture best supports standing meetings in order for this to become a natural part of everyday work among office workers.

## 5. Conclusions

In this explorative study, standing meetings were found to be feasible and highly attractive in the office workplace, and regular participation was related to a reduction in sitting time during work hours. The right facilities were important for standing meetings to be successful. Walking meetings were associated with many barriers and with being difficult to implement, and had no effect on sitting time.

## Figures and Tables

**Figure 1 ijerph-17-01713-f001:**
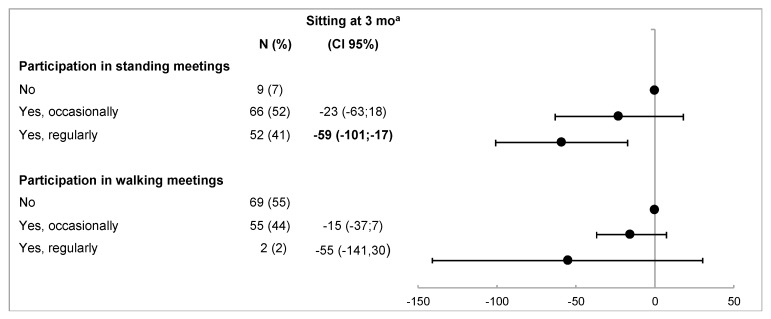
Participation in standing and walking meetings and association with sitting time at 3 months follow-up (*n* = 127). Boldface indicates statistical significance (*p* = 0.006). ^a^ Sitting/8 h workday at 3 months follow-up compared to baseline. The model included participation in standing and walking meetings (exposure); sitting time at 3 months (outcome); and baseline sitting time, age and sex.

**Table 1 ijerph-17-01713-t001:** Participant characteristics at baseline (*n* = 173).

Characteristic	N (%)	Mean (SD)
**Sociodemography and health**		
Women	105 (61)	
Age, years		47 (10)
Tertiary education	130 (76)	
BMI obese (>30)	33 (20)	
Smoker	18 (11)	
Self-rated health excellent/very good	57 (33)	
**Sitting and physical activity ^a^**		
Sitting time, min/8 h workday		345 (54)
Standing time, min/8 h workday		82 (45)
Sitting time, min/8 h leisure		291 (53)
MVPA in leisure, min/8 h leisure		45 (22)
**Facilities for standing and walking meetings ^b^**		
Know where to have standing meetings	144 (94)	
Know where to have walking meetings	135 (89)	
Satisfactory facilities for standing meetings	132 (92)	
Satisfactory facilities for walking meetings	125 (93)	

BMI, Body Mass Index; MVPA, Moderate-to-Vigorous Physical Activity (total time spent walking fast (>100 steps/min), running, climbing stairs, rowing and cycling); SD, Standard Deviation. ^a^ Measured with Actigraph attached on thigh (*n* = 162). ^b^ Questionnaire at 3 months follow-up.

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
