# Peer review of "Standing Meetings Are Feasible and Effective in Reducing Sitting Time among Office Workers—Walking Meetings Are Not: Mixed-Methods Results on the Feasibility and Effectiveness of Active Meetings Based on Data from the “Take a Stand!” Study"

_ijerph, 2020, doi:10.3390/ijerph17051713_

Round 1

Reviewer 1 Report

General Comments

    The purpose of this study was to evaluate the feasibility and effectiveness of standing and walking meetings by using a cluster-randomized trial, consisting of multiple 15 components including the possibility of active meetings. The authors found that standing meetings were feasible and well-liked by the employees, and having frequent standing meetings was associated with reduced sitting time.

Major Concerns

    The most critical defect of this study is not presenting or including control participants. The reviewer could not understand why the authors did not include control participants. It is suspected that a behavioral change of the intervention group might influence that of the control group if this randomization study was conducted in the same office. Anyway, the authors should mention why this study included intervention-offices only (173 participants from 10 clusters).

   However, the authors described as if they analyzed and compared data obtained from the intervention and control groups, for example as the following sentences:

    In abstract, participants, who participated in standing meetings on a regular basis, had 51 min less sitting per 8 h workday (95%CI -92;-9) compared to participants who did not participate in standing meetings at all.

    In 3.2. Effectiveness, Office workers, who participated in standing meetings on a regular basis, had 51 min less sitting time per 8h workday after 3 months (Confidence Interval, CI 95% -95;-9), compared to office workers, who did not.

    As several limitations are stated by the authors as the following, they are all critical and should be improved.

    “Limitations include that participation in standing and walking meetings was measured by a simple self-report question discriminating between no, occasional and regular participation; however, it is unclear how participants understood the terms ‘occasional’ and ‘regular’ and thus we have no information on the actual frequency of participation.”

    “Furthermore, we cannot tell whether the lower sitting time among standing-meeting participants was only due to less time spend sitting in meetings or combined with increased use of their sit-stand-desk.”

Author Response

Response to reviewer 1

General Comments

    The purpose of this study was to evaluate the feasibility and effectiveness of standing and walking meetings by using a cluster-randomized trial, consisting of multiple 15 components including the possibility of active meetings. The authors found that standing meetings were feasible and well-liked by the employees, and having frequent standing meetings was associated with reduced sitting time.

Major Concerns

    The most critical defect of this study is not presenting or including control participants. The reviewer could not understand why the authors did not include control participants. It is suspected that a behavioral change of the intervention group might influence that of the control group if this randomization study was conducted in the same office. Anyway, the authors should mention why this study included intervention-offices only (173 participants from 10 clusters).

Thank you very much for your relevant comments.

We agree that the description of the study population has not been clear. The present study includes participants from intervention clusters only, as they were asked about facilities and participation in standing and walking meetings in the questionnaires. Control participants were asked the most necessary questions, in order not to contaminate or influence them, as they were told to behave as usual.

Randomization was done by cluster and participating offices were separated by walls, floors or locations – and employees within clusters did not collaborate. Thus, risk of contamination from the intervention group was low.

Naturally, it would have been interesting to be able to compare meeting participation between groups and to control for baseline values. This was not possible in the present study, however, based on the promising results of our explorative study, future studies might be designed with increased focus on especially standing meetings.

In order to address the issues regarding the study population, we have improved the materials and methods section as follows:

Specifying that randomization was by clusters (l.76-77):

“Randomization was conducted on the cluster-level 1:1 to intervention or control within each workplace.”

We have added a reason as to why only intervention offices were included (l.81-82):

”We included intervention-offices only (10 clusters with a total of 173 participants), as the necessary information for the present study was obtained only in this group.”

And finally, it has been specified which data was collected in the intervention group (l.88-90):

”Information on frequency of standing and walking meetings, facilities and background variables was collected in the intervention group by web-based questionnaires at baseline and at 3 months follow-up.”

And (l.94-95):

”Interview data were obtained from 11 focus groups with participants and 20 semi-structured interviews with ambassadors and leaders conducted at the 10 intervention offices after the final follow-up at 3 months.”

   However, the authors described as if they analyzed and compared data obtained from the intervention and control groups, for example as the following sentences:

    In abstract, participants, who participated in standing meetings on a regular basis, had 51 min less sitting per 8 h workday (95%CI -92;-9) compared to participants who did not participate in standing meetings at all.

    In 3.2. Effectiveness, Office workers, who participated in standing meetings on a regular basis, had 51 min less sitting time per 8h workday after 3 months (Confidence Interval, CI 95% -95;-9), compared to office workers, who did not.

Thanks for your comment. All results are on the intervention group only, comparing intervention-participants with different level of participation in standing and walking meetings.

In order to clarify this, we have elaborated on the effectiveness-results as follows (l.184-186):

“Office workers, who participated in standing meetings on a regular basis, had 59 min less sitting time per 8h workday after 3 months (Confidence Interval, CI 95% -101;-17), as compared to office workers in the intervention group, who did not participate in standing meetings.”

    As several limitations are stated by the authors as the following, they are all critical and should be improved.

We admit that the study has some relevant limitations, which is primarily due to the explorative design of the study. Take a Stand! was a cluster-randomized trial designed to assess sitting time effects of a multicomponent intervention; however, we thought it would be interesting to explore the potential of active meetings further, as the literature on this topic is quite limited. In order to emphasize the explorative design we have made a number of correction throughout the paper:

In the abstract (l.13):

”The aim of the present study was to explore the feasibility and effectiveness of standing and walking meetings.”

And (l.25):

”This explorative study conclude that, in office workplaces, standing meetings were feasible and well-liked by the employees…”

In the introduction (l.42):

”Thus, the purpose of this study was to explore the feasibility of standing and walking meetings”

In the conclusion (l.250):

”In this explorative study, standing meetings were found to be feasible…”

    “Limitations include that participation in standing and walking meetings was measured by a simple self-report question discriminating between no, occasional and regular participation; however, it is unclear how participants understood the terms ‘occasional’ and ‘regular’ and thus we have no information on the actual frequency of participation.”

We acknowledge this is a very relevant limitation, however, we do not have further information on the exact frequency of meetings as this was not the main outcome of the trial. From interviews and workshop-observations we know that standing meetings were implemented regularly in most offices with varying frequency, and that the length varied from brief status meetings (~10 min) to longer meetings (>30 minutes). Future studies on standing and walking meetings could preferably obtain more detailed information on the frequency and length of these meetings.

To clarify this, the paragraph has been improved as follows (l.227ff):

“Limitations include that participation in standing and walking meetings was measured by a simple self-report question discriminating between no, occasional and regular participation; however, it is unclear how participants understood the terms ‘occasional’ and ‘regular’ and thus we do not know the exact frequency. In addition, it is unknown exactly how long the meetings lasted; however, the included standing meetings were standing-only meetings as another question considered meetings with standing or walking elements.”

“Furthermore, we cannot tell whether the lower sitting time among standing-meeting participants was only due to less time spend sitting in meetings or combined with increased use of their sit-stand-desk.”

We agree, that this is a weakness of the study, however we have overall information only on sitting and standing during the workday from accelerometers and questionnaire information on meeting participation. From interviews and observations, we know that some participants had daily standing meetings; however, for some regular standing meetings were weekly, and even if they are supplemented by ad hoc standing meetings this is probably not enough to achieve the full 59 minutes reduction as observed. Probably, the participants standing regularly at meetings also stand more at their desk compared to participants not standing in meetings. Reasons for this could be that among these participants, there was a higher uptake of the intervention, and that standing at meetings had a positive effect on standing during the rest of the workday. This might be because participants at meetings influence each other positively and because standing meetings remind participants of the project and increase awareness of behaviour change.

In order to elaborate on this the paragraph in the discussion has been extended into the following (l.233ff):

“Furthermore, we cannot tell whether the lower sitting time among standing-meeting-participants was due to less time spend sitting in meetings or combined with increased use of their sit-stand-desk. As the reduction was 59 minutes per day it is reasonable to assume a combination of reduced sitting at meetings and during the rest of the day. It might be that standing meetings contribute to the general awareness of the project and the social support between participants, both of which contributes to reduced sitting during the rest of the day.”

Reviewer 2 Report

This is an interesting study that demonstrates a potentially large impact reducing sedentary time in officer workers through the use of standing meetings in the workplace. The effect size is significant, and given the high number of the population who work in an office environment the population level impact could be substantial.

I think the authors need to provide a clearer justification as to why they have analysed a sub-set of the overall study, choosing only the intervention group and not including any results from the trial. Although I think this approach is defensible, they have not explained why they have taken this approach and what the downside is. 

The potential confounding factor alluded to in the discussion of the use of sit-stand desks needs further discussion- at what point were all the participants given sit/stand desks- was this prior to the introduction of walking meetings? what data is available on how individuals used the sit stand desks- were those who liked standing meetings those that were already active outside the work and who used their desks a lot in stand mode? I am unclear if this data is used in the model- clarity is needed as to what factors were included in the model.

Author Response

Response to reviewer 2

This is an interesting study that demonstrates a potentially large impact reducing sedentary time in officer workers through the use of standing meetings in the workplace. The effect size is significant, and given the high number of the population who work in an office environment the population level impact could be substantial.

I think the authors need to provide a clearer justification as to why they have analysed a sub-set of the overall study, choosing only the intervention group and not including any results from the trial. Although I think this approach is defensible, they have not explained why they have taken this approach and what the downside is. 

Thank you very much for your relevant comments.

We agree that the description of the study population has not been clear. The present study includes participants from intervention clusters only, as they were asked about facilities and participation in standing and walking meetings in the questionnaires. Control participants were asked the most necessary questions, in order not to contaminate or influence them, as they were told to behave as usual.

Randomization was done by cluster and participating offices were separated by walls, floors or locations – and employees within clusters did not collaborate. Thus, risk of contamination from the intervention group was low.

Naturally, it would have been interesting to be able to compare meeting participation between groups and to control for baseline values. This was not possible in the present study, however, based on the promising results of our explorative study, future studies might be designed with increased focus on especially standing meetings.

In order to address the issues regarding the study population, we have improved the materials and methods section as follows:

Specifying that randomization was by clusters (l.76-77):

“Randomization was conducted on the cluster-level 1:1 to intervention or control within each workplace.”

We have added a reason as to why only intervention offices were included (l.81-82):

”We included intervention-offices only (10 clusters with a total of 173 participants), as the necessary information for the present study was obtained only in this group.”

And finally, it has been specified which data was collected in the intervention group (l.88-90):

”Information on frequency of standing and walking meetings, facilities and background variables was collected in the intervention group by web-based questionnaires at baseline and at 3 months follow-up.”

And (l.94-95):

”Interview data were obtained from 11 focus groups with participants and 20 semi-structured interviews with ambassadors and leaders conducted at the 10 intervention offices after the final follow-up at 3 months.”

The potential confounding factor alluded to in the discussion of the use of sit-stand desks needs further discussion- at what point were all the participants given sit/stand desks- was this prior to the introduction of walking meetings? what data is available on how individuals used the sit stand desks- were those who liked standing meetings those that were already active outside the work and who used their desks a lot in stand mode? I am unclear if this data is used in the model-

We agree that this might seem a bit confusing. The explanation is that in Denmark, everyone has a sit-stand desk; however, they are seldomly used. In the intervention, participants were provided with a number of different strategies within four themes to reduce sitting during workhours. One theme was having standing and walking meetings and another was to use the sit-stand desk actively during the day. Thus, it is very likely that participants reduced sitting time in several ways and the total reduction of 59 minutes/8 hour workday is probably due to a combination of standing meetings and standing at the desk. Probably, standing meetings has a positive effect on intervention uptake due to both awareness and increased social support.

Unfortunately, we do not have specific data available on the exact use of sit-stand desk or on the lengths of meetings, as the overall aim of the intervention was reduced sitting time. However, future studies might preferably improve measures in this regard in order to obtain increased knowledge of different strategies to reduce sitting.

The multilevel mixed-effects linear regression included baseline sitting apart from the outcome (sitting at 3 months) and the factor of interest (meeting participation). We did not control for leisure time physical activity, as previous studies regarding subgroups of physical activity levels found no differences in sitting time effect between participants. Furthermore, we found no intervention effect on leisure time sitting or physical activity. Thus, it seems that in the present study leisure time activity and workplace sitting were unrelated. However, we performed sensitivity analysis including one by one baseline values of objectively measured total physical activity and leisure time steps in the model. Results are included as supplementary material; however, they did not change conclusions.

Information on sit-stand desks is given in the materials and methods section (l.80-81):

“All participants had sit-stand desks prior to inclusion.”

The statistical model has been specified in the data analysis section (l.108ff):

”Statistical analysis was conducted using STATA/IC-14.0. Multilevel mixed-effects linear regression was used with 3-months sitting time as outcome, taking baseline sitting, age and sex into account. Variables on participation in standing and walking meetings were tested consecutively.”

and under the figure showing the results of the model (figure 1):

”The model included participation in standing and walking meetings (exposure); sitting time at 3 months (outcome); and baseline sitting time, age and sex.”

Sensitivity analyses have been described (l.111):

”Sensitivity analyses included baseline values of leisure time physical activity and steps one by one.”

And in the results on effectiveness (l.187-189):

“Sensitivity analysis, taking baseline physical activity and steps into account, did not change these results (supplementary file, figure a and b).”

In order to elaborate on this weakness in the discussion the paragraph has been extended into the following (l.233ff):

“Furthermore, we cannot tell whether the lower sitting time among standing-meeting-participants was only due to less time spend sitting in meetings or combined with increased use of their sit-stand-desk. As the reduction was 59 minutes per day it is reasonable to assume a combination of reduced sitting at meetings and during the rest of the day. It might be that standing meetings contribute to the general awareness of the project and the social support between participants, both of which contributes to reduced sitting during the rest of the day.”

clarity is needed as to what factors were included in the model.

We agree, that it was unclear, which factors were included in the statistical model. To make this clear, we conducted new analysis including the following factors in the main model: sitting at work at 3 months (outcome), meetings participation (exposure) and controlling for baseline sitting, age and sex. This has resulted in the following changes in the manuscript:

In the abstract (l.22):

”Participants, who participated in standing meetings on a regular basis, had 59 min less sitting per 8 h workday (95%CI -101;-17)”

In the section on data analysis the model has been specified (l.108-109):

”Statistical analysis was conducted using STATA/IC-14.0. Multilevel mixed-effects linear regression was used with 3-months sitting time as outcome, taking baseline sitting, age and sex into account.”

And in the results section, updating figure 1 and (l.184-185):

“Office workers, who participated in standing meetings on a regular basis, had 59 min less sitting time per 8h workday after 3 months (Confidence Interval, CI 95% -101;-17)”

Round 2

Reviewer 1 Report

The authors should clearly describe that "this study has no control conditions or at least data are not compared with the control conditions" in the limitation section. Eliminate a phrase "controlled trial" throughout the manuscript. 

Author Response

Thanks for your comments.

We agree that this should be clear. We have therefor added the following line in the limitations section (l.206):

“Limitations include the lack of a control group...”

Furthermore, we have minimized the use of ‘controlled trial’ to a few sentences describing the original study (consequently named ‘The Take a Stand!-study’ (l.14 and 50)) and made the following changes:

In the title (l.2):

“Standing meetings are feasible and effective in reducing sitting time among office-workers—walking meetings are not: Mixed-methods results on feasibility and effectiveness of active meetings based on data from Take a Stand!-study”

Omitting ‘randomized controlled trial’ from keywords.

In materials and methods, we have omitted details on randomization and moved the section on intervention group to the beginning of the section (l.68ff):

“Four workplaces (three public and one private) with four to six offices (clusters) participated in Take a Stand! reaching a total of 317 participants from 19 clusters. We included intervention-offices only (10 clusters with a total of 173 participants), as the necessary information for the present study was obtained only in this group. Details on inclusion and exclusion criteria, recruitment process, randomization, and data on included offices are described elsewhere [8].”